
# CodonGenie: optimised ambiguous codon design tools

Neil Swainston[1], Andrew Currin[1], Lucy Green[1], Rainer Breitling[1,2], Philip J. Day[3] and Douglas B. Kell[1,2]

[1] Manchester Centre for Synthetic Biology of Fine and Speciality Chemicals (SYNBIOCHEM), University of Manchester, Manchester, United Kingdom
[2] School of Chemistry, University of Manchester, Manchester, United Kingdom
[3] Faculty of Biology, Medicine and Health, University of Manchester, Manchester, United Kingdom

## ABSTRACT

CodonGenie, freely available from http://codon.synbiochem.co.uk, is a simple web application for designing ambiguous codons to support protein mutagenesis applications. Ambiguous codons are derived from specific heterogeneous nucleotide mixtures, which create sequence degeneracy when synthesised in a DNA library. In directed evolution studies, such codons are carefully selected to encode multiple amino acids. For example, the codon NTN, where the code N denotes a mixture of all four nucleotides, will encode a mixture of phenylalanine, leucine, isoleucine, methionine and valine. Given a user-defined target collection of amino acids matched to an intended host organism, CodonGenie designs and analyses all ambiguous codons that encode the required amino acids. The codons are ranked according to their efficiency in encoding the required amino acids while minimising the inclusion of additional amino acids and stop codons. Organism-specific codon usage is also considered.

## INTRODUCTION

Site-directed mutagenesis of DNA is an established technique of generating libraries of DNA variants in a controlled manner, and has applications in a range of fields, primarily that of protein engineering (*Jäckel, Kast & Hilvert, 2008*), but also in more fundamental research including the study of sequence-to-fitness relationships (*Hietpas, Jensen & Bolon, 2011*). The design of mutant protein libraries typically involves a manual process in which required sites for mutation are selected and ambiguous codons (those containing mixtures of nucleotides) designed to introduce controlled variation in these positions.

In this process, one may wish to design a codon to specify any subset of amino acids in a given position. Since each amino acid may be included in the subset or otherwise, the number of possible subsets is $2^{20}-1$, i.e., there are 1,048,575 possible subsets of 20 amino acids. (Each of the sets can be represented by a 20-digit binary number, where a one at position $n$ indicates that amino acid $n$ is included in the set, and a zero indicates that it is absent. There are $2^{20}$ such numbers, but one of them represents the empty set and is thus not counted here.) Not all of these 1,048,575 subsets of 20 amino acids are uniquely

Corresponding author
Neil Swainston,
neil.swainston@manchester.ac.uk

designable using ambiguous codons, of which there are only 3375. (There are 15 ($=2^4$–1) relevant nucleotide codes ("letters"), ranging from the completely unambiguous A, C, G and T representing a single nucleotide, to the completely ambiguous N representing all 4 nucleotides (*Cornish-Bowden, 1985*). There are $15^3 = 3,375$ triplet codons that can be assembled from this 15-letter alphabet of ambiguous codes, compared to the $4^3 = 64$ codons that can be constructed from the standard 4-letter alphabet of unambiguous nucleotides.)

Given the degeneracy of the codon table, there are often multiple ways to encode a chosen set of amino acids. The experimenter must (a) decide if it is feasible to encode all desired amino acids (*Mena & Daugherty, 2005*); (b) determine whether this creates an acceptable number of sequence combinations (depending on screening capability and throughput) (*Currin et al., 2015*; *Kille et al., 2013*; *Lutz, 2010*; *Pines et al., 2015*); and (c) consider the codon usage of the organism to be used (*Nakamura, Gojobori & Ikemura, 2000*). It therefore follows that the design of ambiguous codons is non-trivial.

CodonGenie is therefore introduced to provide a quick and easy-to-use means of designing optimal ambiguous codons, considering the above parameters according to the user input, and ranking the ambiguous codons with respect to their suitability for expression in a target host organism. The tool is designed to be both human- and computer-readable, providing both a simple web browser interface and a RESTful webservice API.

## MATERIALS & METHODS

### Algorithm

The standard codon table is such that 17 of the 20 naturally occurring amino acids are encoded by codons with fixed bases in the first and second positions, with the third "wobble"-position allowing variation that accounts for the degeneracy of the DNA code. Determining optimal ambiguous codons for combinations of amino acids involves the following process, which is optimized for computational efficiency, compared to a brute-force examination of all possible ambiguous codons:

Align the first two positions and select the most specific ambiguous bases to encode the alignment. For example, with the combination asparagine and isoleucine (encoded by AA[CT] and AT[ACT] respectively), the alignment of the first two positions is A[AT], i.e., AW.

All combinations of aligned wobble positions are calculated, i.e., [CA], [CC], [CT], [TA], [TC], [TT]. These are then collapsed into unique sets, in this example giving [CA], C, [CT], [TA] and T.

The first two and wobble position bases are combined to produce candidate ambiguous codons, which are scored as described below.

Three amino acids (leucine, arginine and serine) cannot be simply encoded by codons with fixed bases in the first and second positions. (For example, both CTN and TT[AG] encode leucine.) For combinations including these more complex residues, the above algorithm is performed for each encoding and the results combined.

Note that CodonGenie returns not only the most "specific" ambiguous codons, that is, the codons that provide the fewest DNA variants whilst encoding all target amino acids. Providing results that include less specific ambiguous codons, which may also encode

additional amino acids, allows the user to perform a trade-off between library size and codon specificity, depending on the experimental objective. A smaller library is generally advantageous for screening purposes, but may contain codons that are unfavoured by the target host organism.

## Scoring

The goal of the scoring scheme is to preferentially rank the most efficient ambiguous codons. That is, the ambiguous codons that encodes all of the required amino acids while minimising the encoding on non-desired amino acids.

The score for an ambiguous codon is therefore defined as the mean of the *value*, $v_i$, of each of the codons that it encodes. For codons that encode required amino acids, $v_i$ is the ratio of the frequency of the codon $f_i$ and the frequency of the most frequent synonymous codon $f_j$ for the amino acid that it encodes. For codons that encode non-required amino acids, $v_i$ is zero.

$$\text{score} = \frac{1}{|C|}\sum_{i \in C} v_i, \quad \text{where}$$

$$v_i = \begin{cases} \dfrac{f_i}{\max\left(\{f_j : j \in S_i\}\right)} & i \in R \\ 0 & i \notin R \end{cases}$$

$C = \{\text{all variants of ambiguous codon } c\}$

$A = \{\text{target amino acids}\}$

$a_i$ : amino acid encoded by codon $i \in C$

$f_i$ : codon usage frequency of codon $i \in C$

$S_i = \{j : a_j = a_i\}$ Set of synonymous codons of codon $i$

$R = \{i \in C : a_i \in A\}$ Set of codon variants of $c$ encoding target amino acids.

This scoring algorithm thus achieves a principled trade-off between codon specificity, library size and codon favourability (according to the codon usage preferences of the target organism).

## Web service access

CodonGenie also offers a RESTful web service interface, supporting its integration with software pipelines. The Design method can be accessed by specifying required amino acids and required host organism (as an NCBI Taxonomy id *Federhen, 2012*) as follows:

http://codon.synbiochem.co.uk/codons?aminoAcids=DE&organism=4932

Similarly, the Analyse method can be accessed by specifying a variant codon and the required organism:

http://codon.synbiochem.co.uk/codons?codon=NSS&organism=4932

CodonGenie also provides web service interfaces for accessing supported organisms. The first allows all organisms to be listed, showing NCBI Taxonomy id and name, and the second allows the collection to be searched according to a given term:

http://codon.synbiochem.co.uk/organisms/

http://codon.synbiochem.co.uk/organisms/escher

In all cases, results are returned in json format.

## Distribution

The web application is freely available from http://codon.synbiochem.co.uk. CodonGenie is written in Python (using the Flask framework) and HTML/Javascript (using the Bootstrap and AngularJS libraries) and is packaged as a Docker application for ease of deployment. Source code is available from https://github.com/synbiochem/CodonGenie.

## RESULTS AND DISCUSSION

CodonGenie provides a simple web interface affording two functions: (a) the design, and (b) the analysis of ambiguous codons. Considering the Design module, the user specifies the combination of amino acids to be encoded and an organism in which the library will be expressed. The codon usage table is automatically extracted from the Codon Usage Database (*Nakamura, Gojobori & Ikemura, 2000*), which as of May 2017 provided support for 35,792 organisms. CodonGenie then calculates suitable ambiguous codons and presents these in an interactive table (see Fig. 1).

The Analyse module provides the functionality of checking an existing ambiguous codon. Users specify a variant codon and required host organism, and the results returned indicate which amino acids are encoded along with their codon usage frequency.

The benefit of CodonGenie can be exemplified by the design of an ambiguous codon to encode non-polar amino acids phenylalanine, leucine, isoleucine, methionine and valine. A simple and widely used ambiguous codon to encode this subset is NTN, which equates to 16 DNA variants. However, CodonGenie identifies that these same amino acids can be encoded by the DTK codon (where D denotes [AGT] and K denotes [GT]) using six variants. Selecting DTK therefore means fewer enzyme variants need to be screened to test all sequence combinations. This benefit is particularly significant when encoding multiple variant codons. For example, when using 3 DTK codons the library size is reduced from 4,096 ($16^3$) to 213 ($6^3$) combinations.

An example of the importance of considering codon usage of the target host organism can be seen when considering the design of an ambiguous codon to encode the set of five non-polar amino acids (F, I, L, M and V) considered above. For *E. coli*, the preferred codon is DTK (ATG |T |GT), with a score of 0.88. DTS (ATG | T | GC) also encodes all five amino acids using 6 variants, but with a score of 0.68. In *Streptomyces coelicolor*—a commonly used host for antibiotic production (*Pickens, Tang & Chooi, 2011*), the ranking is reversed, with DTS being preferred with a score of 0.79, substantially higher than that of 0.29 for DTK. The reason for this can be found in the codon usage frequencies of each of these organisms, as shown in Table 1: The codons DTK and DTS differ by specifying either GT or GC in the third position, respectively. Taking the example of encoding phenylalanine, F, the codon TTT encoded by ambiguous codon DTK is preferred over TTC (encoded by DTS) in *E. coli* by a frequency of 0.64 to 0.36. By contrast, *S. coelicolor* strongly prefers TTC to TTT to encode F, with frequencies of 0.97 to 0.03, respectively. A similar preference is observable in the codon usage frequencies for encoding isoleucine, I, in *S. coelicolor*, where ATC has a frequency of 0.95 compared to that of 0.03 for ATT. Thus, *S. coelicolor* has a strong preference for the variant codon containing C in the "wobble" position, and this is

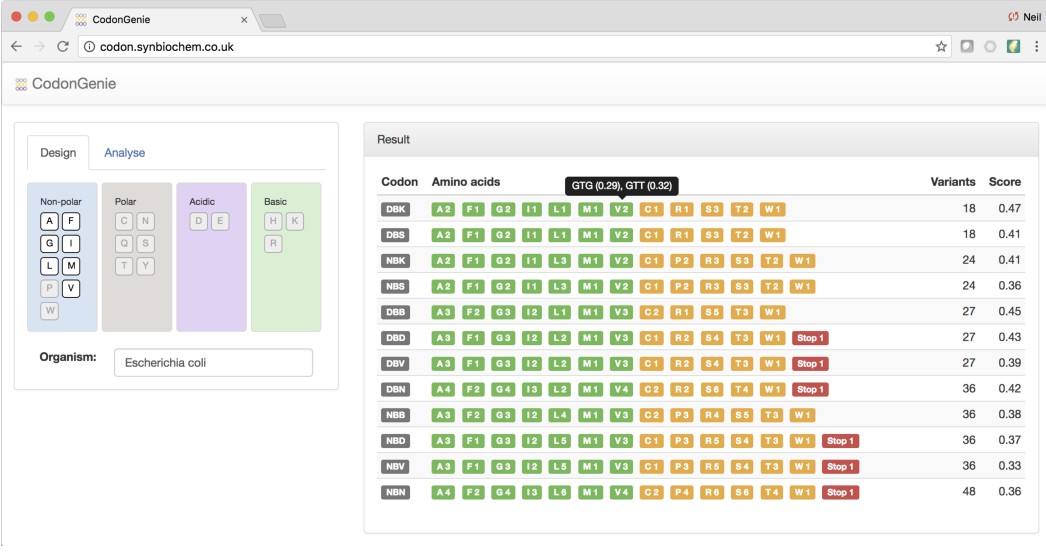

**Figure 1** **CodonGenie Design interface.** Users specify required amino acid combinations in the left-hand side panel. Amino acids are grouped together in the interface in subsets of polar, non-polar, acidic and basic residues. In this example, the non-polar residues A, F, G, I, L, M and V have been selected. Variant codons are listed in the Result panel, ordered by increasing number of Variants and decreasing codon Score (see Methods). The most specific codons are prioritised (e.g., the preferred codon in the above example, DBK, is [AGT][CGT][GT] and therefore encodes 18 DNA variants). Variant codons are shown in grey, with their encodings shown in green, orange and red for required amino acids, additional amino acids and stop codons, respectively. A given variant codon may encode an amino acid multiple times, and this is displayed in the output. For example, the preferred codon DBK encodes valine twice (with GTG and GTT) and these encodings and their codon usage frequencies may be visualised through a tooltip.

reflected in the scores of 0.79 for DTS and 0.29 for DTK. Organism-specific codon usage is therefore a key consideration in the design of ambiguous codons for a given host.

CodonGenie adds to a toolkit of existing software tools for ambiguous codon selection, which includes AA-Calculator (*Firth & Patrick, 2008*) and DYNAMCC (*Halweg-Edwards et al., 2016*). In contrast to AA-Calculator, CodonGenie ranks designed ambiguous codon based on their suitability for use in a given host organism. DYNAMCC also scores designed codons but offers complementary functionality to CodonGenie, as it designs sets of ambiguous codons to encode a set of amino acids with no off-target amino acid encoding and minimal redundancy. CodonGenie designs single ambiguous codons to encode a desired set of amino acids, which may also include off-target amino acids, allowing users to make a conscious trade-off between a larger library and the ease of generating such a library with a single ambiguous codon.

The above example of Table 1 illustrates a key difference between CodonGenie and DYNAMCC. Where CodonGenie will provide a list of individual ambiguous codons that will encode all desired amino acids (and potentially additional, off-target amino acids), DYNAMCC returns a single, best-scoring set of ambiguous codons that encode all desired amino acids with minimal redundancy. In the case of F, I, L, M and V, DYNAMCC returns the set of codons WTT (encoding F and I and L) and VTG (encoding M and V). The advantage of the DYNAMCC approach is in increased efficiency of the library: five DNA variants

**Table 1  Comparison of codon usage frequencies for ambiguous codons encoding F, I, L, M and V in** *Escherichia coli* **and** *Streptomyces coelicolor.* Specific codons from two variant codons DTK and DTS are given, along with their codon usage frequency in the two organisms. For the amino acids F, I and V, there is a preference for codons with T in the third ("wobble") position in *E. coli*, and a preference for C in the wobble position for *S. coelicolor*. This preference is reflected in the differences in scores for the ambiguous codons for the two organisms.

| Amino acid | Codon | Ambiguous codon | Codon usage frequency | |
|---|---|---|---|---|
| | | | *E. coli* | *S. coelicolor* |
| F | TTC | DTS | 0.36 | 0.97 |
| | TTT | DTK | 0.64 | 0.03 |
| I | ATC | DTS | 0.31 | 0.95 |
| | ATT | DTK | 0.47 | 0.03 |
| L | TTG | DTK and DTS | 0.13 | 0.03 |
| M | ATG | DTK and DTS | 1.00 | 1.00 |
| V | GTC | DTS | 0.19 | 0.58 |
| | GTG | DTK and DTS | 0.29 | 0.36 |
| | GTT | DTK | 0.32 | 0.02 |

encode the five desired amino acids, while CodonGenie's solution of DTK or DTS encode six DNA variants, thus producing a larger library. The advantage of CodonGenie's solution lies in the ease in which the library can be produced with a single ambiguous codon.

CodonGenie provides a clean, intuitive web-based user interface which requires minimal user input, and which takes advantage of modern web-application development libraries such as AngularJS and Bootstrap. AngularJS (https://angularjs.org), developed and maintained by Google, provides a framework for the rapid development of modular, testable single-page web applications. Bootstrap (http://getbootstrap.com), initially developed at Twitter, provides a library of reusable user interface "widgets", such as forms, auto-fill boxes, tables, etc. Using freely available yet commercially developed libraries such as these confers a number of advantages: From a development perspective, the libraries are easy to use, are well documented and are thoroughly tested on a range of browsers (including those on mobile phones and tablets) being used perhaps billions of times a day worldwide. More importantly, the user experience is improved through use of well-developed modules that in many cases users have experienced numerous times previously in various other web applications. As a result, CodonGenie can provide a simple, easy-to-use interface that requires no documentation and can run on many platforms with the minimum of development effort.

CodonGenie is designed to follow the concept of "microservices" (*Williams et al., 2016*). Microservice architecture advocates the breaking down of large, monolithic applications into simple, atomic services of limited scope of functionality. By deconstructing large applications or pipelines (such as a DNA design tool) into a collection of independent units (such as a codon design module), the individual microservices can be developed, tested and deployed in isolation, increasing their reliability and reusability. CodonGenie follows this paradigm (the entire application consists of ~700 lines of code) and allows for integration into larger applications by providing a simple computer-readable RESTful

web service API, as well as making itself available as a Docker container (*Belmann et al., 2015*; *Leprevost et al., 2017*), allowing users to easily redeploy their own instantiation on individual computers and services, or cloud-based platforms.

One example of the use of the CodonGenie as a microservice within a larger application is in automating the design of a synthetic DNA sequence to encode a protein sequence generated from a multiple sequence alignment. Consider a multiple sequence alignment of a hypothetic active site of an enzyme:

```
PFDMR
PIAMR
PLHLR
PMNMR
PVHMR
```

The CodonGenie webservice facilitates the writing of a simple script to automate the process of designing a synthetic DNA sequence that captures the variation encoded in this alignment. By iterating through the alignment, the set of amino acids required at each position can be collected ({P} for position 1, {FILMV} for position 2, etc.). These sets can be submitted to the CodonGenie webservice (along with a desired host organism) and a synthetic DNA sequence built up from the highest-scoring ambiguous codon returned. In practice, CodonGenie would produce the following DNA sequence for *E. coli*:

```
CCG|DTK|VMT|MTG|CGT
```

In this example, the first codon (CCG) is not strictly an ambiguous codon, as it contains no ambiguous nucleotides, given that a single amino acid, P, is required in the first position. The codon returned is the therefore the most frequent codon for encoding proline in *E. coli*. The second codon is the optimum codon for encoding F, I, L, M and V, as shown previously.

This example shows the benefit of offering webservice access to the CodonGenie method. While manually designing an optimised DNA sequence for a short alignment such as this is tractable, performing a similar operation on a longer alignment or a number of alignments in a manual fashion would not be feasible. Example code performing this simple operation is available (https://github.com/synbiochem/CodonGenie/blob/master/codon_genie/example/align.py), giving an indication of the ease with which CodonGenie could be incorporated into more comprehensive DNA design pipelines.

## CONCLUSION

CodonGenie provides two simple-to-use yet valuable tools that aid the design of variant protein libraries in mutagenesis and directed evolution studies. Through both its web and web service interfaces, CodonGenie is amenable to future integration with new and existing variant library design software tools (*Swainston et al., 2014*). Its modular and open-source format allows for straightforward adaptation to emerging needs in the synthetic biology community, in particular the consideration of augmented genetic codes and expanded genetic alphabets (*Lajoie et al., 2013*; *Zhang et al., 2017*).

### Funding
Funding was provided by the Biotechnology and Biological Sciences Research Council (BBSRC; http://www.bbsrc.ac.uk) under grant BB/M017702/1, "Centre for Synthetic Biology of Fine and Speciality Chemicals (SYNBIOCHEM)". This is a contribution from the Manchester Centre for Synthetic Biology of Fine and Speciality Chemicals (SYNBIOCHEM). The funders had no role in study design, data collection and analysis, decision to publish, or preparation of the manuscript.

### Grant Disclosures
The following grant information was disclosed by the authors:
Biotechnology and Biological Sciences Research Council: BB/M017702/1.
Manchester Centre for Synthetic Biology of Fine and Speciality Chemicals (SYN-BIOCHEM).

### Competing Interests
The authors declare there are no competing interests.

### Author Contributions
- Neil Swainston conceived and designed the experiments, performed the experiments, analyzed the data, contributed reagents/materials/analysis tools, wrote the paper, prepared figures and/or tables, performed the computation work, reviewed drafts of the paper.
- Andrew Currin conceived and designed the experiments, performed the experiments, analyzed the data, wrote the paper, reviewed drafts of the paper.
- Lucy Green performed the experiments, analyzed the data, wrote the paper, reviewed drafts of the paper.
- Rainer Breitling, Philip J. Day and Douglas B. Kell analyzed the data, wrote the paper, reviewed drafts of the paper.

### Data Availability
Github: https://github.com/synbiochem/CodonGenie.

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
