# Peer review of "CodonGenie: optimised ambiguous codon design tools"

_PeerJ Computer Science, doi:10.7717/peerj-cs.120_

## Round 0.1 · original submission · Minor Revisions

Firstly, I must apologise for the long period of review - the fault was largely mine. You will find three of the four reviewers have commented positively on your manuscript, but R1 considers that the paper in its current form is not suitable for publication, because no research question has been addressed.

From a pure methodological perspective, R1's principle objection is well supported and I urge you to consider rephrasing aspects of the manuscript to make clear :

1) why a new tool for codon set generation is necessary [ e.g. to support enhanced and synthetic coding systems ]

2) how the design of codon genie meets these needs [ e.g. presenting a real world example where codon genie performs in a superior fashion - or makes its capabilities available to other web applications via its rest api ].

3) Additionally, I would recommend you also describe any user interface design decisions that were made during the course of developing the final version of CodonGenie - other web applications for this purpose have a very different visual style, and it would be relevant to report any design principles or usability optimisations that you implemented.

4) R4 points out that directed evolution is just one of the methods used in protein engineering. Please address this in your revision. R3 additionally highlights other methodologies for large scale mutagenesis methods that are perhaps appropriate to be highlighted in this paper.

5) R3 provides detailed revision suggestions in both the general text and description of the method and applicability of the CodonGenie webapp.R4 also points out that more detail concerning the differences between Codon-Genie and Halweg-Edwards et al. are required. Both R2 and R3 suggest the mathematical details require more rigorous explanation - this would be particularly useful for readers less familiar with protein encoding models used by biological organisms. Additionally, R3 suggests it would be useful to provide an example where organism specific coding systems yield different results with CodonGenie's methods.

6) The first step for users employing codon Genie is to enter the target organism. Although this text box provides an autocomplete list, I found the widget somewhat unreliable if arbitrary text is entered. I suggest you provide a link to the list of supported taxons and a short description of the OTU naming conventions supported (e.g. can NCBI ids be provided ?).

7) Thank you for complying with PeerJ CS's requrements regarding the provision of source code. Please ensure that you create a tag for the version you describe in the paper, and additionally, please also revise your readme.txt to include instructions for docker in addition to the google compute engine.

Please address all of these comments. Again, I apologise for the long delay in returning these reviews to you, and look forward to receiving your revised manuscript.

Reviewer 1 ·

Basic reporting

No comment.

Experimental design

This article describes a simple tool for the design of ambiguous codons for encoding multiple amino acids while minimizing undesirable designs. Additionally, using a specific scoring scheme, the tool can score the ambiguous codons for utilizing codons used preferentially in protein coding genes of a target organism.

The PeerJ Computer Science journal only considers research articles, and this manuscript details the implementation of a software tool that uses a brute force method to rank and suggest ambiguous codons to its user. I personally do not see a research objective, either in the form of a hypothesis that is tested, or an algorithm that offers some non-trivial time and/or space complexity. As such, I do not believe there exists a meaningful research question that is pursued.

Validity of the findings

No research question identified.

Additional comments

The manuscript describes an interesting and well implemented tool that can prove useful to the protein design community. It would benefit from pursuing a research question, such as the success of the methodology and scoring scheme described in experiments involving design and evaluation of protein variant libraries.

Reviewer 2 ·

Basic reporting

The paper by Swainston et al. describes CodonGenie, a freely available online tool for designing ambiguous codons to code for defined sets of amino acids. Such a tool is very valuable when constructing protein variant libraries for directed evolution experiments. The ambiguous codons are then ranked according to their efficiency encoding the desired amino acids.
The paper is well written and the language is clear. However, the authors may consider to explain in a bit more detail how they derive the numbers in the sentence in lines 45 – 48.

Experimental design

no comment

Validity of the findings

The authors' approach is to derive suitable ambiguous codons and determine their "quality" is sound.

Additional comments

The web application is very easy and intuitive to use and I would like to commend the authors on their effort to provide a very slimmed down and clean interface that manages to provide the relevant information in a clear and concise manner.

Reviewer 3 ·

Basic reporting

In this short paper, Swainston and colleagues describe a new online tool named CodonGenie, for helping practitioners of directed evolution to design their degenerate codons. Such tools are useful, although as noted by the authors (and in my revisions, below), they are not the first to provide one.

Overall, the basic reporting (writing, figure presentation etc.) is suitable for publication in PeerJ, with the following revisions:

1. The first two paragraphs should be rewritten. Currently, the primary point of them seems to be to advertise/cite a lot of the authors’ previous papers. The first sentence (lines 26-28) makes it sound as though directed evolution is the only way to do protein engineering – better to clarify that directed evolution is one approach of many (e.g. site-directed mutagenesis, de novo design, ancestral sequence reconstruction). Similarly, how do the authors imagine that site-directed mutagenesis (defined by Wikipedia, no less, as “…used to make specific and intentional changes to the DNA sequence of a gene”) is a method for directed evolution (line 28)? Do they mean site-saturation mutagenesis here? What is the point of the second paragraph, in the context of CodonGenie? My (re-)interpretation of the first two paragraphs is, “There are a lot of directed evolution techniques that make use of degenerate codons, including site-saturation mutagenesis and a number of newer methods.” The newer methods aren’t “in contrast” (line 35) to site saturation. And rather than citing four of their own papers, the authors might also consider other large-scale mutagenesis methods such as PFunkel (Firnberg and Ostermeier, PLoS One, 2012) and EMPIRIC (Hietpas et al., PNAS, 2011).

2. The calculations in lines 46-48 should either be explained, or deleted. While I understand where 1,048,575 comes from (line 46), I suspect many experimentalists would not (but may wish to know, so they can perform similar calculations on their own experimental systems). On the other hand, I can’t fathom where 15^3 and 4^3 (line 48) come from.

Experimental design

1. CodonGenie is a really nice little algorithm, but the authors should better explain the gap that it fills. On line 56, they pertinently cite the DYNAMCC algorithm (Halweg-Edwards et al.). In my experience, the other algorithm closest to CodonGenie is AA-Calculator (Firth and Patrick, Nucleic Acids Res., 2008). CodonGenie has clear points of difference to DYNAMCC (which gives you a set of codons with no redundancy, instead of one optimal codon) and AA-Calculator (which gives all degenerate codon possibilities, but no convenient score to choose between them) – point out these differences to the reader!

2. The authors emphasize organism-specific codon design as a key advantage of CodonGenie (e.g. lines 53, 85, 103, 122-124). Can they provide examples where organism-specific codon usage might change the ‘answer’ provided by CodonGenie? Further discussion, and a table or figure, would be justified. There are perhaps half a dozen genetically-tractable species in routine use as hosts for protein engineering and synthetic biology. Could the authors come up with, for example, a case in which one degenerate codon might be best for E. coli, and another for S. cerevisiae? How commonly (or rarely) does the choice of host affect the identity of the top-ranked codon – once in a blue moon, or more often than we might think?

Validity of the findings

There are no data or experiments, however the algorithm and the scoring function are clearly described. The web application appears to be bug free and easy to use.

·

Basic reporting

The article is well written and clear.

Two minor comments:

1) The introduction equates protein engineering with directed evolution. Directed evolution is usually used in protein engineering projects, but it is certainly possible to also just do rational protein design without any diversity generation and screening.

2) It would be good to comment briefly on how the approach described by Halweg-Edwards et al. cited in the article differs from the approach described in this article. This information would aid the reader in deciding which tool to use for a given task.

Experimental design

no comment

Validity of the findings

The results here are a tool that can be used to design variant libraries for protein engineering applications. The method underlying the tool is solid, the web-based tool works and would be expected to provide value for applications. The only data used for the method are publicly available codon usage tables for different organisms.

---

## Round 0.2 · accepted · Accept

The revised manuscript reads well, and addresses all the reviewer's concerns.

I have taken the liberty to suggest some minor edits. Please see the attached PDF. In particular - table 1 needs to be redrawn.